# Regulation of rice root development by a retrotransposon acting as a microRNA sponge

Jungnam Cho*, Jerzy Paszkowski*

The Sainsbury Laboratory, University of Cambridge, Cambridge, United Kingdom

**Abstract** It is well documented that transposable elements (TEs) can regulate the expression of neighbouring genes. However, their ability to act in trans and influence ectopic loci has been reported rarely. We searched in rice transcriptomes for tissue-specific expression of TEs and found them to be regulated developmentally. They often shared sequence homology with co-expressed genes and contained potential microRNA-binding sites, which suggested possible contributions to gene regulation. In fact, we have identified a retrotransposon that is highly transcribed in roots and whose spliced transcript constitutes a target mimic for miR171. miR171 destabilizes mRNAs encoding the root-specific family of SCARECROW-Like transcription factors. We demonstrate that retrotransposon-derived transcripts act as decoys for miR171, triggering its degradation and thus results in the root-specific accumulation of SCARECROW-Like mRNAs. Such transposon-mediated post-transcriptional control of miR171 levels is conserved in diverse rice species.

DOI: https://doi.org/10.7554/eLife.30038.001

## Introduction

Transposable elements (TEs) constitute a large fraction of eukaryotic genomes. Given their mutagenic potential and largely unknown functions, they were often considered as genomic parasites that are silenced by host epigenetic mechanisms (*Fultz et al., 2015*; *Girard and Hannon, 2008*). However, there is increasing evidence that TEs contribute to various chromosomal functions, to the evolution of genomes by increasing genetic variation, and to the direct regulation of genes (*Lisch, 2013*). Several studies have revealed that TEs in plants endow genes with both coding and regulatory sequences (*Lisch, 2013*). For example, the Arabidopsis transcription factors FHY3 and FAR1, involved in light signalling, are derived from the transposase of the Mutator-like DNA transposon (*Hudson et al., 2003*). The domestication of hAT and Mutator-like transposases contributed to the evolution of the *DAYSLEEPER* and *MUSTANG* gene families, respectively. *DAYSLEEPER* was shown to play a critical role in plant development (*Bundock and Hooykaas, 2005*; *Cowan et al., 2005*; *Knip et al., 2013*; *Knip et al., 2012*). More recently, a protein derived from the transposase of the Pif/Harbinger transposon family was shown to be an inhibitor of POLYCOMB REPRESSIVE COMPLEX 2 (*Liang et al., 2015*).

TEs residing outside protein-coding regions of genes can influence their expression by interfering with promoters, providing enhancers, or altering RNA processing and/or epigenetic regulation. For example, TEs residing in introns or UTRs may alter the availability of splicing sites and/or splicing efficiencies. They can also shift polyadenylation signals or supply binding sites for miRNA and RNA-binding proteins (*Feschotte, 2008*).

In contrast to the numerous examples of local influence on gene regulation in cis, examples of TEs mediating the regulation of distant genes are rare. For example, the Arabidopsis *ddm1* mutant, which is impaired in epigenetic suppression of transposon-derived transcription, accumulates 21-nt small RNAs derived from *Athila* retrotransposons. These small RNAs impair the levels and the

*For correspondence:
jungnam.cho@slcu.cam.ac.uk (JC);
jerzy.paszkowski@slcu.cam.ac.uk
(JP)

**Competing interests:** The authors declare that no competing interests exist.

**eLife digest** An organism's genome contains all of the DNA the individual needs to survive and grow. Transposons are pieces of DNA that can move around the genome. They make up almost half of human DNA and over 85% of the DNA of major crop plants like maize, barley and wheat.

When transposons move they can cause harmful changes in the regions where they insert into the DNA and so cells have mechanisms in place to tightly control the activities of the transposons. However, some transposons cause changes to DNA that are beneficial to the organism. Thus, the relationship between transposons and their host organisms is an example of a delicate but mostly peaceful coexistence. Although the cellular mechanisms controlling transposons are quite well known, the extent to which the transposons affect the ability of organisms to survive, develop and reproduce is poorly understood.

A family of proteins known as the SCARECROW-like transcription factors are important for the roots of plants to develop properly. In other organs such as the leaves or flowers these proteins can cause developmental defects, so the plants carefully control where the proteins are made. Thus, plants normally produce a molecule called miR171 in leaves and flowers, but not in roots, that inhibits protein production by binding to and destabilising the RNA molecules that act as templates to make these proteins.

Cho and Paszkowski have now identified a transposon that produces an RNA molecule with similarities to the RNA templates used to make the SCARECROW-like transcription factors. The experiments show that this transposon RNA is found in very high amounts in roots and mimics the transcription factor RNA so well that miR171 binds to it. This inactivates miR171 in roots to allow the SCARECROW-like transcription factors to be produced.

These findings reveal a new mechanism by which transposons may regulate how plants develop and provide possible new approaches for boosting the growth of rice and other crop plants. Similar regulatory interactions between transposons and their host DNA may also be present in animals and other organisms.

DOI: https://doi.org/10.7554/eLife.30038.002

translation of the mRNA of *OLIGOURIDYLATE BINDING PROTEIN1* (*UBP1*) activated by abiotic stress (*McCue et al., 2012*). Interestingly, although more than 20 genes in Arabidopsis have putative binding sites for transposon-derived small RNAs that would allow regulation analogous to that of *UBP1*, such a network of interactions has not been documented so far (*McCue et al., 2013*).

TE transcripts often contain features resembling micro RNA (miRNA) genes (*Li et al., 2011*) or sequences that are potential targets of miRNAs (*Creasey et al., 2014*). miRNAs are a class of small non-coding RNAs that, directed by their sequences, selectively repress gene expression by translation inhibition or cleavage of mRNA (*Rogers and Chen, 2013*). Interestingly, miRNAs can interact both with their target mRNAs and also with other RNAs containing similar binding sites. Such 'rival' RNAs, which were seen as competing endogenous RNAs (ceRNAs) (*Kartha and Subramanian, 2014*; *Salmena et al., 2011*; *Tay et al., 2014*), can be derived from pseudogenes or long non-coding RNAs and also from protein-coding mRNAs. They may also appear in the form of circular RNAs. Although studies of their transgenic overexpression support activity as miRNA sponges, a possible biological role has not been demonstrated so far by their loss of function (*Thomson and Dinger, 2016*).

Although, transposon-derived transcripts were not considered previously to be a source of ceRNAs, we decided to search rice transcriptomes for indications of this activity. Approximately 35% of the rice genome consists of TEs, which is significantly higher than in Arabidopsis (14%) (*International Rice Genome Sequencing Project, 2005*; *Arabidopsis Genome Initiative, 2000*). More important and similar to maize (*Erhard et al., 2009*; *Hollick et al., 2005*; *Parkinson et al., 2007*), rice mutants impaired in epigenetic silencing of transposons, such as *DICER-LIKE 4* (*DCL4*) or *RNA-DEPENDENT RNA POLYMERASE 6* (*RDR6*), show severe developmental abnormalities (*Liu et al., 2007*; *Song et al., 2012*), whilst the corresponding mutants of Arabidopsis have no apparent morphological aberrations (*Xie et al., 2005*). This dissimilarity suggests that restrained TE-

derived transcription is important for rice development (*Liu et al., 2007*; *Song et al., 2012*; *Wei et al., 2014*).

Here we have specifically investigated TE-derived transcripts as potential regulators of rice development. We found that numerous TEs display patterns of transcriptional activity that are associated with particular plant tissues. Remarkably, a significant proportion of TE-derived transcripts correlate with the mRNA levels of genes transcribed in the same tissues and the two classes of transcripts often share patches of homology. Notably, the sequences of these patches appear to be significantly enriched for miRNA-binding sites. Therefore, we investigated whether some of the transposon-derived transcripts act as ceRNAs. Experiments to test this hypothesis led to the identification of a novel domesticated retrotransposon that is highly expressed in rice roots and that acts as a ceRNA post-transcriptionally controlling the level of miR171. This particular ceRNA is also a target mimic of miR171, which potentially enhances its sponging activity towards miR171. Tissue-specific adjustment of miR171 levels is essential to the proper development of roots and this appears to be regulated by a retrotransposon-derived ceRNA. We demonstrated that mutations in its miR171-binding site result in an abnormal root system.

## Results

### Predicted interaction of transposon-derived RNAs with host miRNAs

To examine tissue-specific abundance of TE-derived transcripts in rice, we accessed publicly available RNA sequencing (RNA-seq) datasets for various tissues of rice (*Figure 1A*). We considered only the datasets of Japonica rice, cultivar Nipponbare and applied the same data-processing pipeline to raw sequencing results generated in different laboratories (details in the Materials and methods). This way we achieved consistent results and samples representing particular tissues were clustered together (*Figure 1A*). Such combined dataset yielded 2961 transcribed TEs (filtered for maximal RPKM (Reads Per Kilobase per Million reads)>1). Remarkably, the TEs were transcribed in most rice tissues and their transcriptomes exhibit clear tissue specificity (*Figure 1A* and *Figure 1—figure supplement 1A*). The rice expression patterns differ from those of Arabidopsis, where TEs are activated in a non-selective way and only in seed endosperm and the vegetative cells of pollen grains (*Figure 1—figure supplement 1B*) (*Slotkin et al., 2009*). Thus, in rice, the two-dimensional correlation matrix of TE transcriptomes showed distinct TE groups reflecting their RNA abundance in various tissues and at different developmental stages (*Figure 1—figure supplement 1A*). In contrast, Arabidopsis TEs exhibit more uniform expression patterns (*Figure 1—figure supplement 1B*).

We detected TE-derived transcripts in rice tissues that do not contribute to the germ line (e.g. endosperm, leaves and roots). These transposon activities, even when resulting in insertions, would not be transmitted to the next generation. In the case of such apparently unproductive TE activity, reactivated TEs may possibly be regulatory or, as in Arabidopsis, may be the RNA substrates of small RNAs involved in TEs silencing (*Creasey et al., 2014*), or may simply reflect insignificant transcriptional noise.

For regulatory activity that influences gene expression, we assumed that TE transcripts would share homologies with the transcripts of co-expressed genes. Indeed, multiple alignments revealed homology patches in approximately 64% of co-transcribed TEs, while silent TEs matched only by 41% (*Figure 1—figure supplement 1C*). Moreover, sequence comparison of expressed TEs and corresponding genes showed a strong bias towards sense direction, by which the aligned regions of TEs were much longer than for TEs matching gene transcript in the antisense direction (*Figure 1—figure supplement 1D*, left panel). Intriguingly, orientation bias was observed when the alignments were sought against processed mRNAs of genes but not when introns were included (*Figure 1—figure supplement 1D*).

Next, we sought sequence features differentially enriched in the co-expressed TEs and found enrichment for miRNA-binding sites (*Figure 1B*). This raised the possibility that some TE transcripts interfere with miRNA-mediated gene regulation, possibly by competing for miRNA binding. As two different RNAs interacting with the same miRNA would need to be co-expressed in the same tissue, we compared tissue-specific expression of TEs and matching genes. We found 763 and 400 TE-gene pairs in sense and antisense orientation, respectively, including 282 sense and 111 antisense pairs with correlation coefficient above 0.5 (*Figure 1C*). Such correlated expression patterns between

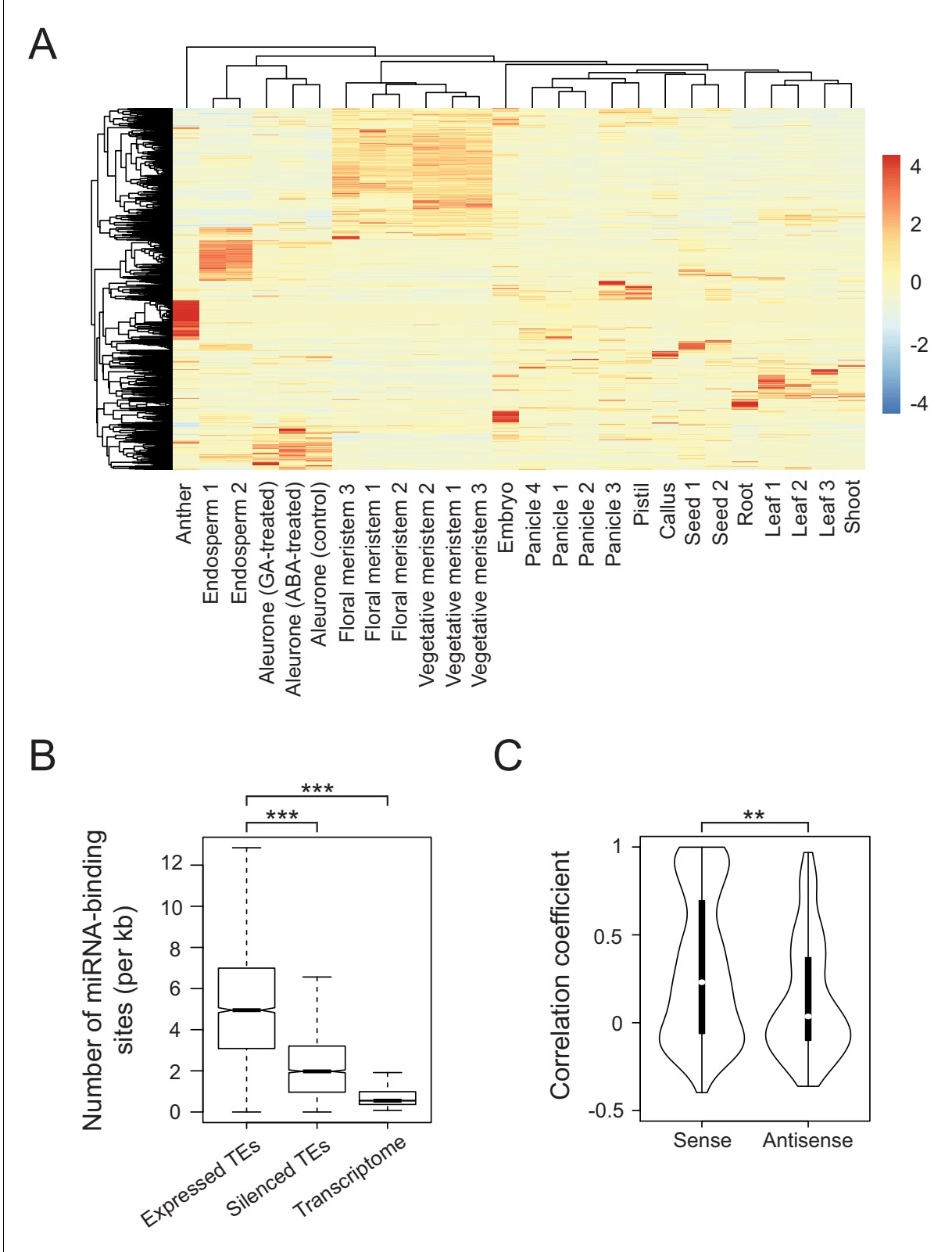

**Figure 1.** Developmental control and potential miRNA target mimicry of TEs in rice. (**A**) TE expression patterns in various rice tissues. The numbers indicate biological replicates (for endosperm and meristem) or the independent datasets of the same tissues. (**B**) The number of predicted miRNA-binding sites corrected for the lengths of transcripts. miRNA target sites were predicted by psRNATarget with default settings (http://plantgrn.noble.org/psRNATarget/). The asterisks indicate statistical differences determined by the Wilcoxon rank sum test. ***p<e-10. (**C**) The violin plot for the

*Figure 1 continued on next page*

*Figure 1 continued*

Pearson's correlation coefficient between TEs and matching gene expression patterns. TE-gene pairs sharing miRNA-binding sites are separated into sense and antisense matching. **p<e-05.

DOI: https://doi.org/10.7554/eLife.30038.003

The following figure supplement is available for figure 1:

**Figure supplement 1.** Tissue-specific expression patterns and sequence alignments of TEs in rice.

DOI: https://doi.org/10.7554/eLife.30038.004

mRNAs of genes and TE-derived transcripts in sense orientation was most evident in roots (*Figure 1—figure supplement 1E*). Collectively, the results of our examination of tissue-specific transcriptomes are consistent with the hypothesis that TEs regulate gene expression by miRNA sequestration.

## MIKKI is a root-specific domesticated retrotransposon

To test this hypothesis, we rigorously re-analysed 61 root-specific rice transcriptomes and selected a particular TE, which we named *MIKKI* ('decoy' in Korean), for further investigation (*Figure 2A*).

First, we validated RNA-seq results of root-specific transcription of *MIKKI* by RT-qPCR (*Figure 2B*). To distinguish the spliced transcript from precursor mRNA (pre-mRNA), we designed primers across exon junction or within the intron, respectively (*Figure 2B*, left and right panel). The RT-qPCR results confirmed that the mature *MIKKI* transcript is highly abundant in roots, present at low levels in leaves, and almost absent in panicles (*Figure 2B*, left panel). A similar expression pattern was observed for unspliced RNA but at much lower levels (*Figure 2B*, right panel). These results are consistent with tissue-specific regulation of *MIKKI* at the transcriptional level.

*MIKKI* is a TE-derived locus which includes *Osr29* Long Terminal Repeat (LTR) retrotransposon. Based on sequence divergence between the two LTRs, an *Osr29* element transposed about 3.7 million years ago (mya, *Figure 2C* and *Figure 2—figure supplement 1A*). We also found sequences of three further retrotransposons, *BAJIE*, *Osr30* and *Osr34*, inserted subsequently into *Osr29* (*Figure 2C*). Advanced degeneracy prevented estimate of the insertion times of *Osr30-* and *Osr34-* related sequences; however, the generation time of the solo LTR derived from the *BAJIE* family was estimated to be approximately 1.2 mya (*Figure 2C* and *Figure 2—figure supplement 1B*). The *MIKKI* gene product was predicted to encode just a partial reverse-transcriptase (RTase) protein and no other protein domains were found (*Figure 2—figure supplement 1C*). Given that several amino acid residues essential for catalytic activity of RTase are mutated in MIKKI's RTase (*Figure 2—figure supplement 1D*), it seems unlikely that MIKKI's RTase domain would be active. Thus, we concluded that *MIKKI* is not expected to have regulatory role at the protein level. Most important, the mature transcript of such a rearranged *Osr29* (*MIKKI*) was found to contain an imperfect binding site for miR171, generated by a splicing event joining *BAJIE* solo LTR sequences to specific sequences of *Osr29* (*Figure 2C,E* and *Figure 2—figure supplement 1E*). miR171 is one of the miRNAs conserved across the plant kingdom and previous studies revealed that Arabidopsis miR171 (ath-miR171) is abundant in flowers but sparse in roots (*Figure 2—figure supplement 2C–E*; *Llave et al., 2002*). Rice miR171 (osa-miR171) displays a similar expression pattern (*Figure 2D*, left panel and *Figure 2—figure supplement 2B*). Thus, miR171 levels seem to be similar in particular tissues of these two distant species, highest in reproductive organs and lowest in roots.

It is well documented that ath-miR171 targets mRNAs encoding SCARECROW-Like (SCL) transcription factors for cleavage and, thus, *SCL* transcript levels display patterns opposite to miR171 (*Llave et al., 2002*). The same *SCL* transcript distribution was observed in rice (*Figure 2D*, right panel), implying the regulation of *SCL* transcript stability also by osa-miR171. Moreover, the sequence identity of rice and Arabidopsis *SCL* mRNAs across the miR171-binding region is also consistent with the evolutionary conservation of miR171-mediated cleavage of *SCL* transcripts (*Figure 2—figure supplement 2F*). Indeed, analyses of the RNA degradome in rice panicles (*Wu et al., 2009*) revealed specific cleavage of *OsSCL21* mRNAs at the osa-miR171 binding region (*Figure 2E*, left panel). We also examined whether the *MIKKI* transcript is also targeted by osa-miR171 but found no signals indicative of site-directed cleavage of *MIKKI* transcripts at the putative miR171-binding site (*Figure 2E*, right panel). Importantly, there are two mismatches in the miR171-binding region of

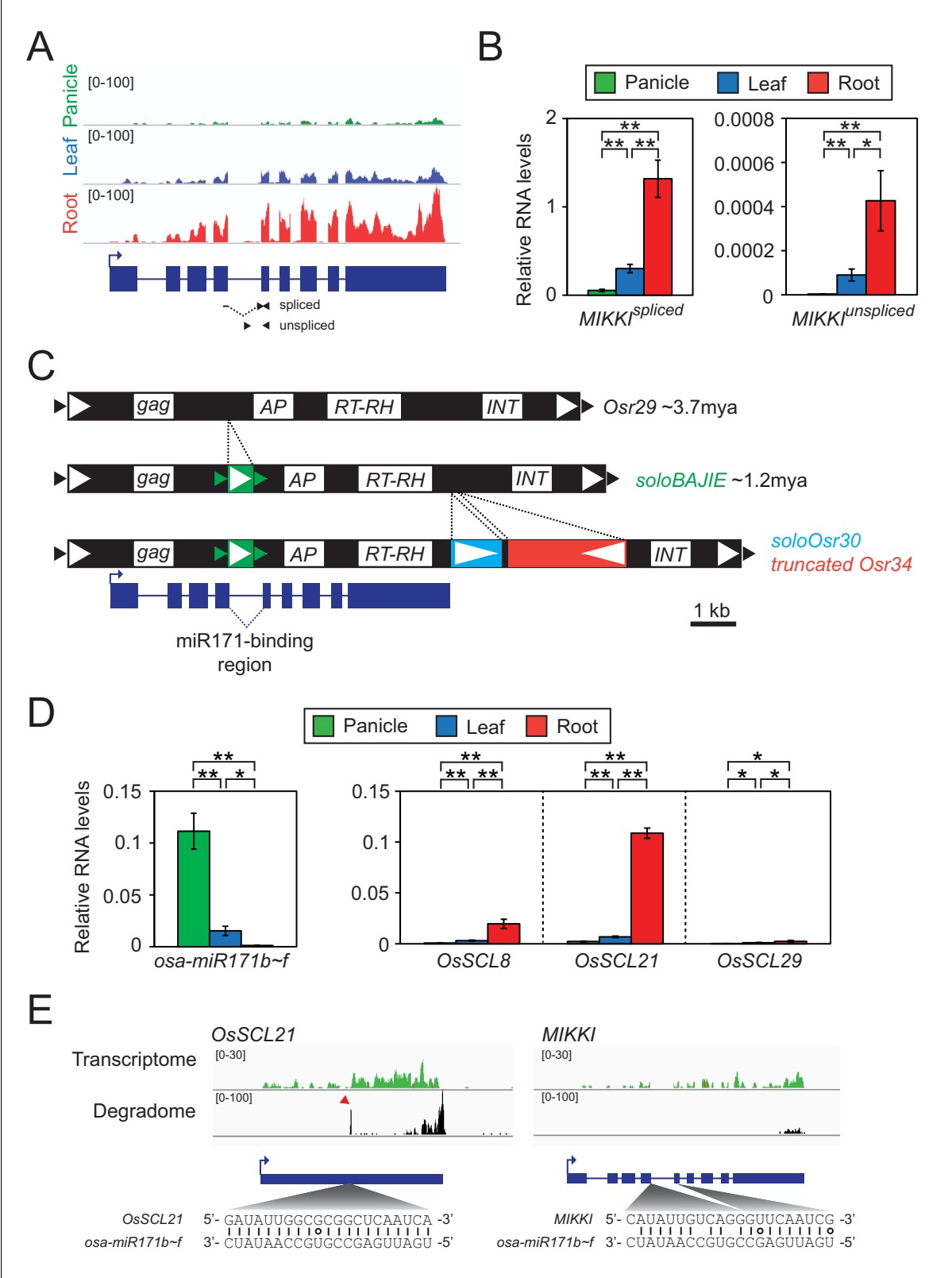

**Figure 2.** Root-specific *MIKKI* transcripts may act as target mimics for miR171. (**A**) Top, the root-specific expression pattern of *MIKKI* shown as a snapshot of the RNA-seq genome browser. Bottom, structure of a *MIKKI* transcript; blue boxes and lines represent exons and introns, respectively. The arrow indicates the transcription start site and the primers used in (**B**) are indicated as arrowheads. The primer spanning splice junction is shown as a dashed line. (**B**) *MIKKI* expression pattern revealed by RT-qPCR. Relative levels of spliced and unspliced *MIKKI* mRNA in the left and right panels,

*Figure 2 continued on next page*

*Figure 2 continued*

respectively. Data are presented as mean ± standard deviation (sd) of three biological replicates performed in technical triplicate. The asterisks indicate statistical differences determined by Student's t-test. **p<0.005; *p<0.05. (C) Schematic diagram of evolution of *MIKKI* locus. The open and closed arrowheads are the long terminal repeat (LTR) regions and target site duplications, respectively. Different families of retrotransposons are presented by the different colours marked on the right, together with their estimated ages. AP, aspartyl protease; RT-RH, reverse transcriptase-RNaseH; INT, integrase. Intron 4 is shown as a dashed line. (D) Levels of osa-miR171b ~ f and *OsSCLs* in different tissues as determined by RT-qPCR. Error bars represent mean ± sd of three biological replicates performed in technical triplicate. The asterisks indicate statistical differences determined by Student's t-test. **p<0.005; *p<0.05. (E) Transcriptome and degradome data from rice panicles showing the *OsSCL21* (left) and *MIKKI* (right) loci. The base pairing of osa-miR171 to *OsSCL21* and *MIKKI* is shown below. The red arrowhead indicates the peak of cleaved end sequences of *OsSCL21* mRNA. Watson-Crick and Wobble base-pairing between osa-miR171b ~ f and *OsSCL21* or *MIKKI* are indicated as lines and circles, respectively.
DOI: https://doi.org/10.7554/eLife.30038.005

The following figure supplements are available for figure 2:

**Figure supplement 1.** Sequence alignment of *MIKKI*-associated LTRs.
DOI: https://doi.org/10.7554/eLife.30038.006
**Figure supplement 2.** Conservation of expression patterns and target sequences of miR171.
DOI: https://doi.org/10.7554/eLife.30038.007

*MIKKI* at positions 11th and 14th. Conservation of nucleotides at these sites is known to be essential for target RNA cleavage (*Jeong et al., 2013*; *Liu et al., 2014*; *Llave et al., 2002*). It is, therefore, possible that the mismatches around the cleavage sites in *MIKKI* transcripts attenuate the cleavage activity of osa-miR171. Altogether, these data are consistent with the possibility that *MIKKI* is a target mimic of osa-miR171 in rice roots.

## MIKKI acts as a target mimic of osa-miR171

To examine the target mimicry of the *MIKKI* transcript towards miR171, we overexpressed *MIKKI* in both rice and Arabidopsis (*Figure 3—figure supplement 1A and B*, top panel), and applied RT-qPCR analyses. miR171 levels were downregulated in independent transgenic lines generated from both plant species (*Figure 3A*, top panel and *Figure 3—figure supplement 1B*, middle panel) and the transcript levels of target genes were markedly upregulated (*Figure 3A*, bottom panel and *Figure 3—figure supplement 1B*, bottom panel). Previous studies revealed abnormalities in floral organs of Arabidopsis plants in which ath-miR171 levels were decreased by overexpression of artificial target mimics (*Ivashuta et al., 2011*; *Todesco et al., 2010*). Consistent with these observations, plants ectopically overproducing *MIKKI* transcripts also displayed severe defects in reproductive organs and low fertility (*Figure 3B,C* and *Figure 3—figure supplement 1C*).

To address directly the developmental role of the *MIKKI* retrotransposon, we generated the *MIKKI* mutants *mikki-1* and *mikki-2* using CRISPR-Cas9 (*Miao et al., 2013*). To ensure the targeting specificity of guide RNAs (gRNA), we designed them to target the unique junction region between *Osr29* and *BAJIE* (*Figure 3—figure supplement 2A*). Transgenic plants were examined by sequencing for mutations in this region and two independent alleles were found (*Figure 3—figure supplement 2A and B*). The *mikki-1* allele had a 2 bp deletion at the splice donor site that resulted in retention of the intron. Intron retention disrupted the miR171-binding site and generated multiple premature stop codons (*Figure 3—figure supplement 2A*). This transcript is likely recognized by a nonsense-mediated mRNA decay pathway and rapidly turned over (*Shoemaker and Green, 2012*). Indeed, the RT-qPCR analyses revealed thousand-fold reduction of *MIKKI* transcripts in the *mikki-1* mutant (*Figure 3—figure supplement 2C*). The *mikki-2* allele has an 8 bp deletion in the region containing the miR171-binding site (*Figure 3—figure supplement 2A*). This deletion did not alter RNA levels but was predicted to lose target recognition by osa-miR171 (*Figure 3—figure supplement 2A and C*).

Next, we performed RT-qPCR on the wildtype (wt) and the mutants. The levels of osa-miR171 were high in both *mikki-1* and *mikki-2* (*Figure 3D*, top panel and *Figure 3—figure supplement 2E*). This correlated with a decrease in RNA levels of *OsSCL21* targeted by osa-miR171 (*Figure 3D*, bottom panel). In Arabidopsis, mutation of *SCLs* leads to defects in root development (*Wang et al., 2010*). From a Korean rice seed bank we obtained two independent mutant alleles of *OsSCL21* that showed the highest transcript levels among *OsSCLs* targeted by miR171 (*Figure 3—figure supplement 3*). Similar to Arabidopsis, the roots of both *osscl21* mutants were shorter than wt (*Figure 3—*

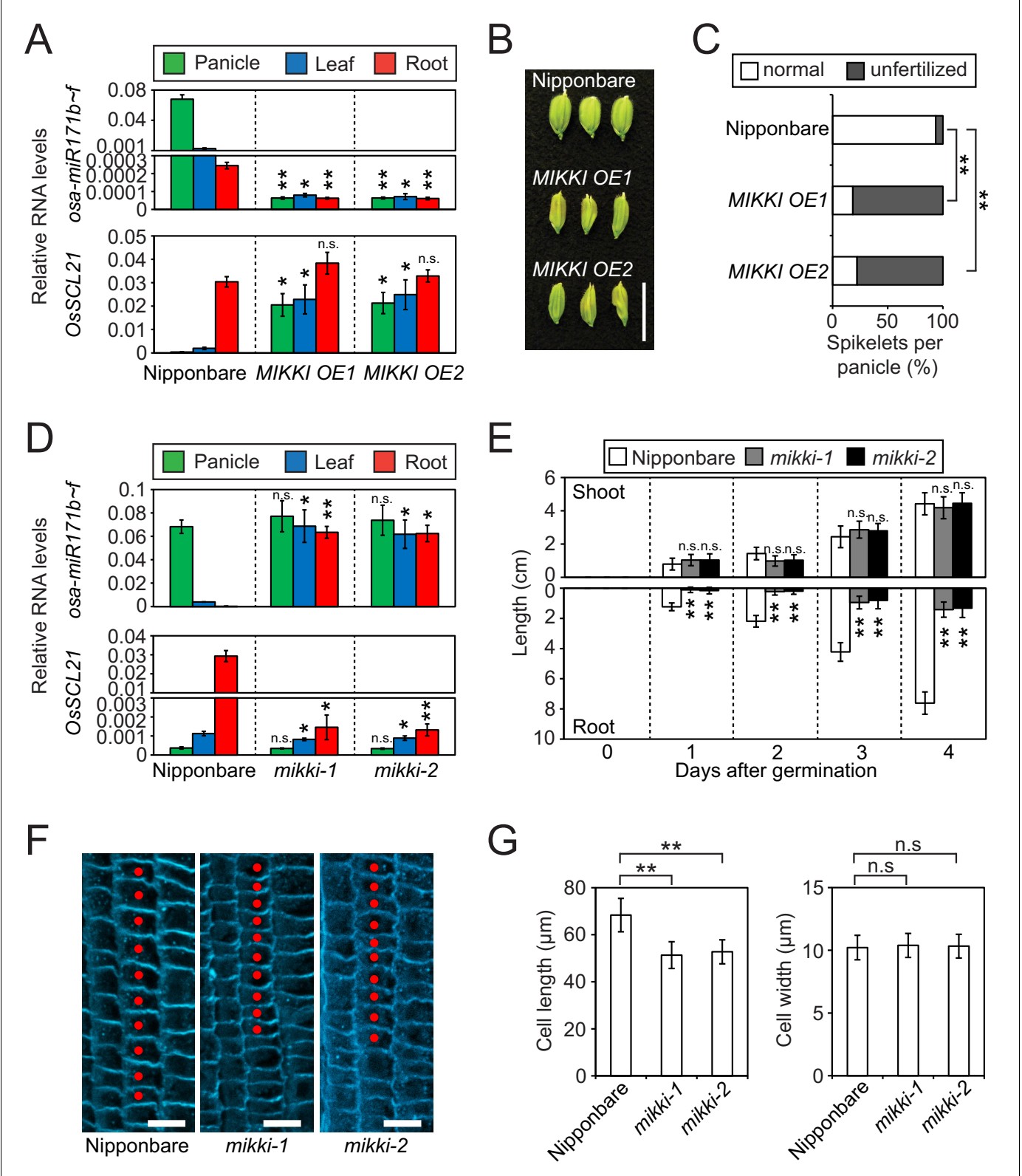

**Figure 3.** *MIKKI* negatively regulates the level of miR171. (**A**) Repression of osa-miR171b ~ f (top) and derepression of its target gene (bottom) in *MIKKI* overexpression lines. RNA was extracted from panicles, leaves and roots. Error bars represent mean ± sd of three biological replicates performed in technical triplicate. The asterisks indicate statistical differences in comparison to the same tissues of wildtype (wt) determined by Student's t-test. \*\*p<e-10; \*p<e-05; n.s., not significant. Wt Nipponbare was segregated from hemizygous overexpressor lines. (**B**) Abnormal spikelet development in

*Figure 3 continued on next page*

*Figure 3 continued*

*MIKKI*-overexpressing lines. Bar = 1 cm. (**C**) Percentage of unfertilized spikelets in overexpression lines. Data are mean of 10 panicles for each genotype. The asterisks indicate statistical differences determined by Student's t-test. \*\*p<e-10. (**D**) Derepression of osa-miR171b ~ f (top) and repression of the target gene (bottom) in *mikki* mutant plants. RNA was extracted from panicles, leaves and roots. Error bars represent mean ± sd of three biological replicates performed in technical triplicate. Wt Nipponbare was segregated from heterozygous mutant plants. The asterisks indicate statistical differences in comparison to the same tissues of wt determined by Student's t-test. \*\*p<e-10; \*p<e-05; n.s., not significant. (**E**) Shoot and root length of wt and the mutants. Data are presented as mean ± sd; n = 15. The asterisks indicate statistical differences in comparison to the same tissues of wt determined by Student's t-test. \*\*p<e-10; n.s., not significant. (**F**) Confocal microscopy images of meristematic regions of wt and the mutants. fifth cortical layer was chosen for comparison. 10 consecutive cells below transition point of meristematic to elongation zone are indicated as red dots. Bar indicates 10 µm. (**G**) Comparison of cell length (left) and width (right) between wt and the mutants. Data are presented as mean ± sd; n = 15. The asterisks indicate significant statistical differences as determined by Student's t-test. \*\*p<e-10; n.s., not significant.

DOI: https://doi.org/10.7554/eLife.30038.008

The following figure supplements are available for figure 3:

**Figure supplement 1.** *MIKKI* overexpression.
DOI: https://doi.org/10.7554/eLife.30038.009

**Figure supplement 2.** *MIKKI* mutation.
DOI: https://doi.org/10.7554/eLife.30038.010

**Figure supplement 3.** Identification of *osscl21* mutant plants.
DOI: https://doi.org/10.7554/eLife.30038.011

*figure supplement 2D*). Subsequently, we examined the development of *mikki-1* and *mikki-2* roots. Root lengths were affected in both mutants, resembling mutants in the *OsSCL21* gene (*Figure 3E* and *Figure 3—figure supplement 2D*). Histological analyses were also performed to observe the cellular consequences of *MIKKI* mutation. Both mutants showed reduced cell elongation above meristematic region, while the cell widths were similar to wt (*Figure 3F and G*). These data are consistent with the hypothesis that *MIKKI* negatively regulates osa-miR171 levels in rice roots, acting through a ceRNA containing target mimic site for osa-miR171.

Next, we asked whether post-transcriptional regulation by a ceRNA with target mimicry is the major regulatory mechanism governing tissue-specific levels of osa-miR171. For this, we determined the levels of the primary transcript of osa-miR171 (pri-osa-miR171) in *MIKKI* overexpression and mutant plants (*Figure 4A,B* and *Figure 4—figure supplement 1A*). The abundance of pri-osa-miR171 was similar in different rice tissues and was not affected by the alteration of *MIKKI* transcript levels or mutation, implying that mature osa-miR171 is regulated post-transcriptionally by the activity of *MIKKI*.

## Species-specific regulation of miR171 level

In contrast to rice, the tissue-specific distribution of primary transcripts of miR171 in Arabidopsis was the same as the mature miRNA, which is consistent with transcriptional regulation and thus an entirely different regulatory mechanism (*Figure 4C*).

*MIKKI* is present and has a conserved structure in AA-genome *Oryza* species (*Figure 4—figure supplement 2A and B*), suggesting strong selective advantage of this particular transposon. *MIKKI* is present in the genomes of *Oryza sativa ssp. indica, O. rufipogon, O. nivara, O. barthii,* and *O. glaberrima* (*Figure 4—figure supplement 2A*). Furthermore, insertion of the *BAJIE*-derived solo LTR and the resulting intron with a miR171-binding site at the splice junction are perfectly conserved (*Figure 4—figure supplement 2B*), implying that the formation of a splicing-dependent miR171 binding site retained in these related species. We examined *MIKKI* splicing in five of these species using available RNA-seq data (*Zhai et al., 2013*; *Zhang et al., 2016*; *Zhang et al., 2014*) and detected identical splicing patterns of the critical intron 4 (*Figure 4—figure supplement 3A*). Moreover, we found that the *MIKKI* homolog of Indica rice displays a developmental expression pattern similar to Japonica rice (*Figure 4—figure supplement 3B*).

We also examined tissue-specific levels of primary transcripts and mature miR171 in monocotyledonous Brachypodium (*Figure 4—figure supplement 1B and C*). As in rice, the primary transcripts of miR171 were high in all tissues examined, suggesting analogous post-transcriptional control of miR171 levels. However, so far we have not identified an *MIKKI*-related element in the genome of Brachypodium.

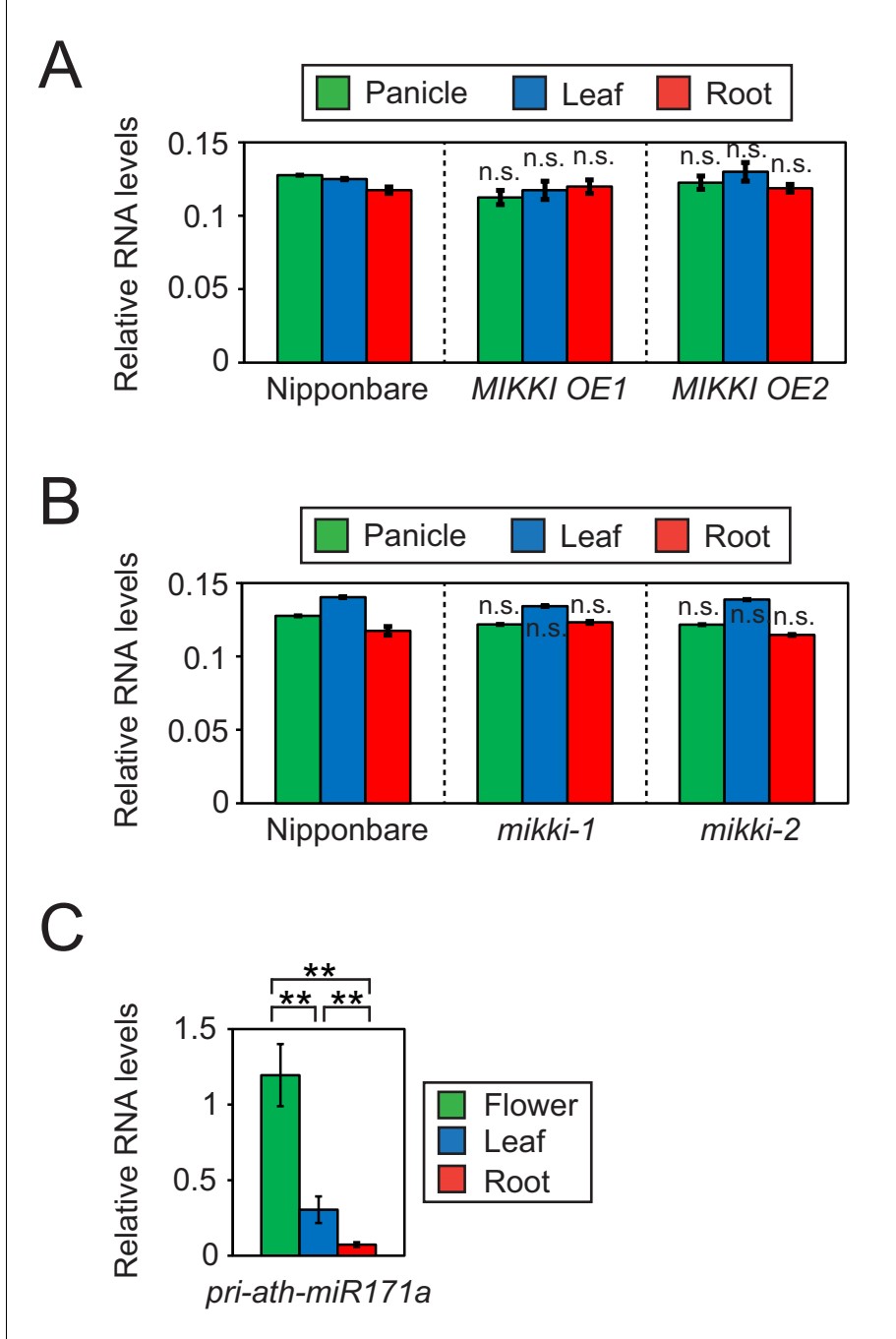

**Figure 4.** Differences in the regulation of miR171 levels in rice and Arabidopsis. (**A and B**) RT-qPCR of the primary precursor of miR171 in wt, overexpressor (**A**) and mutant (**B**) of *MIKKI*. The transcripts levels were normalized to *eEF1α*. Error bars represent mean ± sd of three biological replicates performed in technical triplicate. The asterisks indicate statistical differences in comparison to the same tissues of wt determined by Student's t-test. n.s., not significant. (**C**) RT-qPCR of the primary precursor of miR171a in Col-0 arabidopsis plants. The levels were normalized to *UBQ10* and error bars represent mean ± sd of three biological replicates performed in technical triplicate. The asterisks indicate statistical differences determined by Student's t-test. **p<0.005.

DOI: https://doi.org/10.7554/eLife.30038.012

The following figure supplements are available for figure 4:

**Figure supplement 1.** (**A**) RT-qPCR of primary transcript levels of different miR171 loci in rice.

DOI: https://doi.org/10.7554/eLife.30038.013

**Figure supplement 2.** Conservation of *MIKKI* within the *Oryza* genus.

*Figure 4 continued on next page*

*Figure 4 continued*

DOI: https://doi.org/10.7554/eLife.30038.014

**Figure supplement 3.** Conserved pattern of *MIKKI* expression in the *Oryza* genus.

DOI: https://doi.org/10.7554/eLife.30038.015

## Epigenetic regulation of MIKKI

Since the transcription of TEs is usually controlled by epigenetic mechanisms, we examined DNA methylation and selected histone modifications associated with *MIKKI* in different rice tissues (*Figure 5*). In roots, where *MIKKI* is actively transcribed, its upstream region was enriched in lysine 4 tri-methylation of histone H3 (H3K4me3) and depleted of suppressive H3K9me2 (*Figure 5B*). DNA methylation levels were also lower than in panicles (*Figure 5C*). Analysis on the public RNA-seq data

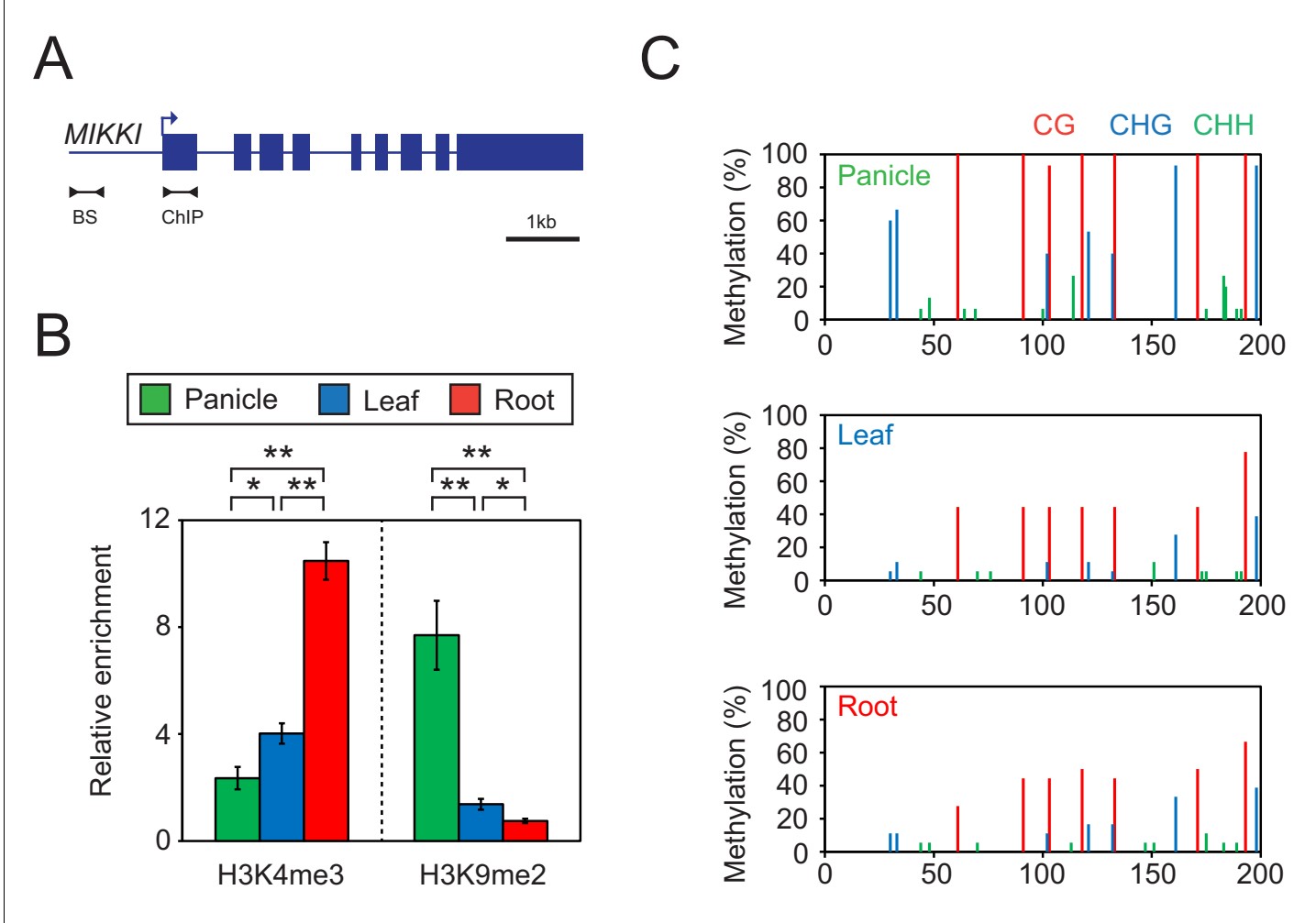

**Figure 5.** Tissue-specific epigenetic signatures of the *MIKKI* locus. (**A**) *MIKKI* structure showing primer positions. BS, bisulfite sequencing; ChIP, chromatin immunoprecipitation. (**B**) H3K4me3 and H3K9me2 levels determined by ChIP-qPCR assay. The amount of immunoprecipitated DNA was normalized to the input levels with *eEF1α* as the internal control region. Error bars represent mean ± sd of three biological replicates performed in technical triplicate. The asterisks indicate significant statistical differences determined by Student's t-test. **p<0.005; *p<0.05. (**C**) DNA methylation levels shown as percent methylation in three different sequence contexts.

DOI: https://doi.org/10.7554/eLife.30038.016

The following figure supplement is available for figure 5:

**Figure supplement 1.** *MIKKI* RNA levels in *osdcl3a* RNAi knock-down lines.

DOI: https://doi.org/10.7554/eLife.30038.017

generated from rice *OsDCL3a* RNAi knock-down lines also showed derepression of *MIKKI* (*Figure 5—figure supplement 1*). These epigenetic signatures were well correlated with tissue-specific transcription of *MIKKI*.

In summary, we propose a model by which *MIKKI* influences rice root development via the regulation of osa-miR171 levels by tissue-specific expression of a ceRNA encoding target mimicry of miR171 (*Figure 6*).

## Discussion

In Arabidopsis, transposable elements are mostly silenced by epigenetic mechanisms preventing their transcription during development of the sporophyte. In contrast, in plant species such as maize or rice, transposon-derived transcripts are detected during specific developmental transitions or in various organs (*Li et al., 2010*; *Tamaki et al., 2015*). Since most of the tissues examined do not contribute to the formation of gametophytes and thus to transgenerational inheritance, the benefits of transposon transcription remain unclear. The prevalent view is that their transcripts are a source of mobile small RNAs that, if transported into germline progenitor cells, would contribute to silencing of transposons there and thus prevent their transgenerational accumulation (*Calarco et al., 2012*; *Creasey et al., 2014*; *Slotkin et al., 2009*). An alternative explanation for the developmental regulation of transposon-derived transcription, however, is that their transcripts have particular functions in a specific tissue or organ. To examine this latter possibility, we systematically analysed tissue-specific transcriptomes of rice transposons by re-analysing available raw RNA sequencing data. These analyses uncovered a surprisingly high fraction of TE-derived transcripts in specific tissues or developmental stages of rice plants. We also observed that transposon-derived transcripts are enriched in

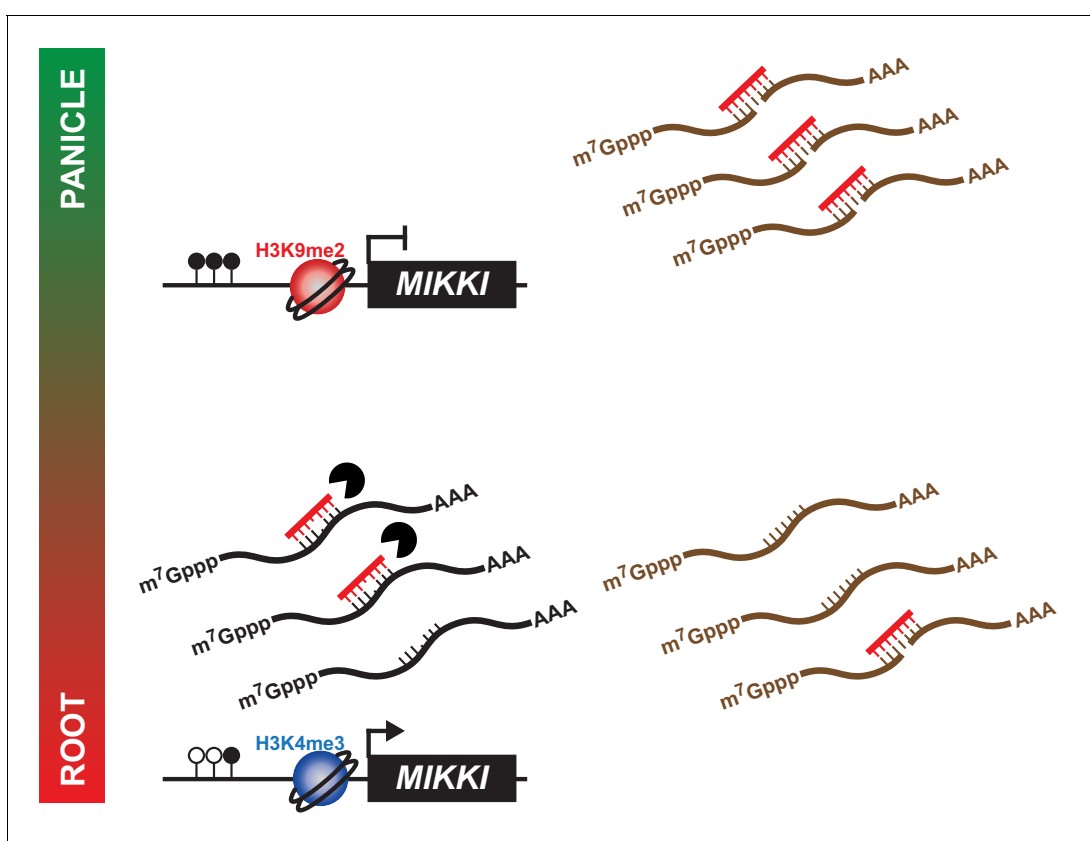

**Figure 6.** A proposed model for the role of *MIKKI* in osa-miR171 control. In rice panicles, *MIKKI* is epigenetically silenced by DNA methylation and repressive histone modifications (shown as closed circles and red sphere, respectively). In roots, *MIKKI* transcripts interact with osa-miR171b ~ f leading to its turnover and stabilization of *OsSCL* mRNAs.
DOI: https://doi.org/10.7554/eLife.30038.018

miRNA binding sites. It has been proposed that Arabidopsis miRNAs trigger generation of transposon-derived small RNAs that contribute subsequently to transposon silencing. These small RNAs may later spread to other cell types (*Creasey et al., 2014*; *Martínez et al., 2016*). This scenario is consistent with the transposon defence hypothesis described above and similar mechanisms could certainly operate in rice.

Alternatively, transposon-derived RNAs containing binding sites for miRNAs could also act as ceRNAs and there are experimental examples in Arabidopsis supporting this possibility. The first and the physiologically important example of an Arabidopsis ceRNA was non-coding RNA *INDUCED BY PHOSPHATE STARVATION 1* (*IPS1*, *Franco-Zorrilla et al., 2007*). The *IPS1* locus is transcriptionally activated upon phosphate starvation and encodes RNA that binds to miR399. miR399 also targets the mRNA *PHO2* gene that encodes an E2 ubiquitin-conjugating-like enzyme affecting the phosphate content of shoots. Importantly, the miR399-binding region in *IPS1* RNA has a 3-nt bulge in the cleavage site and, similar to *MIKKI*, it is resistant to cleavage. This unproductive binding of miR399 to *IPS1* RNA triggers miRNA degradation (*Yan et al., 2012*), thus reducing its level. The mechanism of such a miRNA decoy was termed 'target mimicry'. It has been suggested that the small RNA-specific nucleases in Arabidopsis reduce miRNA levels when target mimics are overexpressed (*Ramachandran and Chen, 2008*; *Yan et al., 2012*). However, the mechanism of miRNA degradation has not been fully elucidated and it is not known how miRNA is recognized for degradation when associated with RNA encoding target mimics but not when associated with mRNAs encoding its bona fide targets.

Subsequently, considerable efforts have been made to identify miRNA target mimics in genomes and transcriptomes of plants (*Fan et al., 2015*; *Meng et al., 2012*) and mammals (*Clark et al., 2014*; *Helwak and Tollervey, 2014*; *Imig et al., 2015*). In addition, further examples of their biological activities have received experimental support (*Franco-Zorrilla et al., 2007*; *Wang et al., 2015*); H.-J. *Wu et al., 2013*). Notably, experimental support for the activity of ceRNAs, including those from plants containing 'target mimicry' sites, is based on transgenic overexpression of micro RNAs or transcripts containing their target mimics. Unfortunately, these assays significantly alter the natural stoichiometry of such regulatory systems. As a consequence, the role of ceRNAs in nature has been queried, given that their relatively low abundance may be insufficient to significantly alter the levels of very dynamically regulated miRNAs (*Thomson and Dinger, 2016*). In addition, mathematical modelling of miRNA target competition supports the notion that target mimics and miRNAs must be at particular levels for maximal effect of target mimicry (*Bosia et al., 2013*; *Figliuzzi et al., 2013*; *Yip et al., 2014*; *Yuan et al., 2015*). Since *MIKKI* transcripts in rice roots are in ample excess over *OsSCL* mRNAs (*Figure 2B and D*), the proportions of the components of the *MIKKI*-miR171-*OsSCLs* module appear to be naturally sufficient for highly effective regulatory activity. Moreover, we have directly examined the biological role of *MIKKI* mRNA as a ceRNA containing target mimics of osa-miR171 by using site-directed mutation of its miRNA binding site or mutations in the splicing site that result in its depletion. These experiments, circumventing artificial changes in the stoichiometry of the interacting components, revealed a regulatory function of *MIKKI* transcripts in the proper development of rice roots.

The tissue-specific posttranscriptional control of miR171 in rice contrasts with restriction of miR171 availability in Arabidopsis roots, which seems to be determined by transcription of miR171. Activation of miR171 is thought to be by AtSCL proteins binding directly to promoters (*Xue et al., 2014*). Given that the promoter sequences of miRNA-encoding genes are generally less conserved than miRNA-coding regions (*Zhu et al., 2015*), corresponding genes in different plant species may be the subject of different transcriptional regulation. The levels of miR171 in different plant tissues are decisive for plant development and posttranscriptional control, implemented by a domesticated retrotransposon, reinforces differential organ distribution of mature miR171. This mechanism seems to be highly conserved among distantly related rice species, suggesting a strong selective advantage. The fact that osa-miR171 and *MIKKI* transcript levels were not affected by *OsSCL21* mutation (*Figure 3—figure supplement 3D and E*) further supports the hypothesis of independent and unique post-transcriptional controlling mechanism emerged. Interestingly, miR171 in the model grass Brachypodium also seems to be regulated posttranscriptionally but a potential ceRNA has not yet been identified.

It has been stated frequently that transposition bursts make a large contribution to host plant genome structure and function (*Lisch, 2013*). However, examples of transposon-mediated control of

plant development through the regulation of distantly located genes have not been reported. Although TEs are an ample source of miRNAs and can potentially act as potent target mimic, they have not been examined for such function. Our study provides the first example of the TE-derived target mimic and thus a novel mechanism for TEs acting as trans-acting regulators of genes. However, for an effective target mimic several conditions should be met, including high transcript levels, good binding affinity and advantageous stoichiometry to miRNA target genes, therefore it is currently still difficult to predict how general this regulatory mechanism will be. We have discovered many more putative TE target mimics in the rice genome (listed in *Supplementary file 1*). However, genetic interference with these hypothetical regulatory loops is difficult due to multicopy components and thus genetic redundancy, as is the case for most transposons. Potentially, the increasingly efficient site-specific alteration of genomes by CRISPR-Cas9 or a population genetics approach using natural variation may improve functional accessibility to the transposon-derived fractions of plant genomes and reveal the extent of their regulatory input.

## Materials and methods

### Plant material and growth conditions

Husks of rice seeds were removed and the seeds were surface sterilized in 20% bleach and germinated in ½-strength Murashige and Skoog media. The 2-week-old seedlings were transferred to soil and grown to maturity in a greenhouse. Root and leaf samples were harvested from 2-week-old seedlings and panicles collected immediately after heading. The rice strains used in this study were *Oryza sativa ssp. japonica cv. Nipponbare*, *O. sativa ssp. japonica cv. Hwayoung* and *O. sativa ssp. indica cv. IR64*. The mutant lines of *osscl21* were identified from a T-DNA tagged population established at Kyunghee University, Korea and genotyped for the selection of homozygotes. (http://cbi.khu.ac.kr/RISD_DB.html/).

### Next generation sequencing analysis

For rice transcriptome analysis, raw FASTQ files of the following RNA sequencing datasets were downloaded: GSE16631, DRA000385, SRP008821, DRA002310, SRP028376 and GSE50778. The adapter-trimmed clean reads were mapped to the reference genome of MSU7 using TOPHAT 2.0, with most of the options set to the default but with some optimization (-g 300). Cufflinks was used to call the RPKM. All the downstream analyses and plotting, for example, heat-maps of transcript levels and correlation matrix, box/violin plots, and statistical analysis were performed in R studio. The transcriptome data of different *Oryza* species were obtained from PRJNA264484, PRJNA264480, PRJNA264485, SRP070627, GSE41797 and analysed as described above. Arabidopsis transcriptome data were obtained from PRJNA314076 and analysed as for the rice transcriptome using the TAIR10 reference genome.

For the degradome and small RNA analysis, raw FASTQ files from GSE18251 and GSE16350 were downloaded and mapped uniquely to the MSU7 reference genome using BOWTIE2. The resulting BAM file was visualized by IGV. Arabidopsis small RNA-seq data were obtained from GSE28591 and analysed as for the degradome. HTSeq was used to calculate the read counts for each miRNA.

### RT-qPCR

Total RNA was extracted using the RNeasy Plant mini kit (Qiagen, Hilden, Germany) following the manufacturer's recommendations. Reverse transcription reactions were performed using the VILO RT kit (Invitrogen, California, USA) with a random hexamer for priming. Real-time quantitative PCR was carried out using the Roche Light-Cycler (Roche, Basel, Switzerland) in a volume of 10 μl and analysed by the ΔΔCt method. All data in this study are the average of three biological replicates performed in technical triplicate ± standard deviation and normalized against *eEF1α*. An Ncode miRNA first-strand cDNA synthesis kit (Invitrogen, California, USA) was used for miRNA quantification; normalization was against miR166, which is expressed constitutively in rice (*Figure 2—figure supplement 2B*). Sequences of primers used are listed in *Supplementary file 2*.

## Small RNA northern blot analysis

A total of 15 µg of RNA was separated on 15% urea-TBE gels (Thermo Fisher Scientific, Massachusetts, USA), transferred to Hybond N + nylon membranes (Amersham Biosciences, Buckinghamshire, UK) and fixed chemically using EDC (Sigma-Aldrich, Missouri, USA). The membranes were prehybridized for 1 hr and hybridized for at least 16 hr in DIG Easy Hyb buffer (Roche, Basel, Switzerland) at 37°C. Membranes were washed twice with 2X SSC (saline sodium citrate), 0.1% SDS. Immunological detection of DIG-labeled probe was performed using DIG wash and block buffer set (Roche, Basel, Switzerland) and DIG luminescent detection kit (Roche, Basel, Switzerland). Luminescent signal was detected with Amersham Imager 600 (Amersham Biosciences, Buckinghamshire, UK).

## Histological analyses of rice roots

Roots from 4-day-old rice plants were fixed in FAA (formaldehyde, acetic acid, ethanol) solution overnight in cold room, wax embedded using Leica ASP300 tissue processor (Leica Biosystems, Wetzlar, Germany) and sectioned by 4 µm using microtome (Leica Biosystems, Wetzlar, Germany). After dewaxing and ethanol washing, samples were stained with Calcofluor White (Sigma-Aldrich, Missouri, USA) to visualize cell wall. Images were taken with a Zeiss LSM 700 confocal microscope (Leica Biosystems, Wetzlar, Germany).

## Chromatin immunoprecipitation

Leaf and root samples were collected from *Oryza sativa ssp. japonica cv. Nipponbare* plants grown for 2 weeks under short-day conditions (10 hr light/14 hr dark). Panicles were harvested immediately after heading from plants grown in the greenhouse. Samples were crosslinked with 1% formaldehyde, flash-frozen, and ground in liquid nitrogen. Chromatin was fragmented by sonication and immunoprecipitated using the following antibodies: H3K4me3 (ab8580; abcam, Cambridge, UK) and H3K9me2 (ab1220; abcam, Cambridge, UK). The immunoprecipitated DNA was quantified by qPCR and normalized against levels of input and the reference genes indicated. All the results are presented as means ± standard deviation (sd) of three biological replicates performed in technical triplicate. Sequences of primers used are listed in *Supplementary file 2*.

## Bisulfite sequencing

Genomic DNA was extracted from rice tissues using the DNeasy plant mini kit (Qiagen, Hilden, Germany). The Epitect bisulfite kit (Qiagen, Hilden, Germany) was used for bisulfite conversion of unmethylated cytosines. The primer design and data analysis used kismet (*Gruntman et al., 2008*). At least 15 clones from each sample were analysed. Sequences of primers used are listed in *Supplementary file 2*.

## Generation of overexpression lines

A cDNA fragment of *MIKKI* from the start to the stop codon was amplified, cloned into the pUN1901 and pGPTVII binary vectors, and transformed into rice and Arabidopsis, respectively (*Walter et al., 2004*; *Wang et al., 2004*). For rice transformation, embryo-derived 2-week-old calli were immersed in agrobacterium-containing media. Transgenic rice plants were obtained after antibiotic selection and differentiation of plantlets. The detailed procedure was as described previously (*Nishimura et al., 2006*). Arabidopsis transformation was by the floral dip method as described previously (*Clough and Bent, 1998*).

## Targeted mutagenesis

The oligonucleotide of the designed guide RNA was inserted into the pOs-sgRNA entry vector and shuttled to the pH-Ubi-cas9-7 destination vector by the LR recombination reaction. The resulting binary vector was transformed into rice as described above (*Miao et al., 2013*). To detect mutation by CRISPR-Cas9, the region containing the targeted region from genomic DNA was amplified, cloned into the pGEM T-easy vector (Promega, Wisconsin, USA) and sequenced. Selected mutant lines were cultured to the next generation to segregate away the T-DNA and individuals homozygous for mutant allele were selected. Sequences of primers used are listed in *Supplementary file 2*.

## Phylogenetic analysis of *MIKKI*

Genome sequences of the selected *Oryza* species were obtained from Ensembl Plants (http://plants. ensembl.org/). Local BLAST analysis was performed manually using the *MIKKI* genomic sequence, followed by multiple sequence alignment in ClustalW2 and visualization by FigTree v.1.4.2 and box-shade v.3.21.

## Age estimation of LTRs

LTR retrotransposon age was estimated as described previously (*Ma and Bennetzen, 2004*). Briefly, for *Osr29*, the divergence was calculated from sequence degeneracy of two LTRs. Age of insertion was computed using the equation: $T = D/2 t$, where $T$ is the time since insertion, $D$ is the divergence and $t$ is the substitution rate of $1.3 \times 10^{-8}$ per site per year as proposed previously (*Ma and Bennetzen, 2004*). For *BAJIE* solo LTR, the sequence was compared to the consensus of *BAJIE* LTR sequences. We assumed that the consensus *BAJIE* LTR sequence represents the youngest copy and used the equation of $T = D/t$.

## Gene accessions

*MIKKI*, LOC_Os06g02304; *OsSCL8*, LOC_Os02g44360; *OsSCL21*, LOC_Os04g46860; *OsSCL29*, LOC_Os06g01620.

## Acknowledgements

We thank Drs. Zhengming Wang and Weibing Yang for technical support and advice on small RNA blots and histological analyses. This work was supported by European Research Council (EVOBREED) [322621]; Gatsby Fellowship [AT3273/GLE].

## Additional information

### Funding

| Funder | Grant reference number | Author |
| --- | --- | --- |
| European Research Council | 322621 | Jungnam Cho Jerzy Paszkowski |
| Gatsby Charitable Foundation | AT3273/GLE | Jerzy Paszkowski |

The funders had no role in study design, data collection and interpretation, or the decision to submit the work for publication.

### Author contributions

Jungnam Cho, Conceptualization, Data curation, Formal analysis, Validation, Investigation, Visualization, Writing—original draft, Writing—review and editing; Jerzy Paszkowski, Conceptualization, Resources, Data curation, Formal analysis, Supervision, Funding acquisition, Validation, Investigation, Visualization, Writing—original draft, Project administration, Writing—review and editing

### Author ORCIDs

Jungnam Cho (iD) https://orcid.org/0000-0002-4078-7763
Jerzy Paszkowski (iD) http://orcid.org/0000-0002-1378-5666

### Decision letter and Author response

Decision letter https://doi.org/10.7554/eLife.30038.053
Author response https://doi.org/10.7554/eLife.30038.054

# Additional files

## Supplementary files

• Supplementary file 1. List of putative TE target mimics in rice. TE-gene pairs with potential miRNA competition were selected based on the following criteria: (1) TEs with significant expression levels (RPKM >1); (2) Sequence matching in sense orientation; (3) miRNA-binding sites within the matching regions; (4) Correlated expression patterns.

DOI: https://doi.org/10.7554/eLife.30038.019

• Supplementary file 2. Sequences of primers used in this study.

DOI: https://doi.org/10.7554/eLife.30038.020

• Supplementary file 3. Tissues selected for Arabidopsis TE coexpression analysis.

DOI: https://doi.org/10.7554/eLife.30038.021

• Transparent reporting form

DOI: https://doi.org/10.7554/eLife.30038.022

## Major datasets

The following previously published datasets were used:

| Author(s) | Year | Dataset title | Dataset URL | Database, license, and accessibility information |
|---|---|---|---|---|
| Guo G, Zhang G, Hu X, Li Q, Zhuang R, Tian W, Huang Q, He Z, Tao Y | 2010 | rice whole transcriptome surveyed by RNA-Seq and Paired-end technology | https://www.ncbi.nlm.nih.gov/geo/query/acc.cgi?acc=GSE16631 | Publicly available at the NCBI Gene Expression Omnibus (accession no: GSE16631) |
| Sakai H, Mizuno H, Kawahara Y, Wakimoto H, Ikawa H, Kawahigashi H, Kanamori H, Matsumoto T, Itoh T, Gaut BS | 2011 | Expression divergence of the rice retrogenes | https://trace.ddbj.nig.ac.jp/DRASearch/submission?acc=DRA000385 | Publicly available at the DNA Data Bank of Japan (accession no: DRA000385) |
| Buell R, Jiang N, Lin H, Davidson R, Gowda M, Hamilton J, Vaillancourt B | 2012 | Comparative transcriptomics of three Poaceae species reveals patterns of gene expression evolution | http://trace.ddbj.nig.ac.jp/DRASearch/study?acc=SRP008821 | Publicly available at the DNA Data Bank of Japan (accession no: SRP008821) |
| Tamaki S, Tsuji H, Matsumoto A, Fujita A, Shimatani A, Terada R, Sakamoto T, Kurata T, Shimamoto K | 2015 | Florigen-induced Transposon Silencing in the Shoot Apex during Floral Induction in Rice | https://trace.ddbj.nig.ac.jp/DRASearch/submission?acc=DRA002310 | Publicly available at the DNA Data Bank of Japan (accession no: DRA002310) |
| Shen J | 2013 | RNA-sequencing Reveals Previously Unannotated Protein-coding and miRNA-coding Genes Expressed in Aleurone Cells of Rice Seed | https://trace.ncbi.nlm.nih.gov/Traces/sra/?study=SRP028376 | Publicly available at the NCBI Gene Expression Omnibus (accession no: SRP028376) |
| Wei L, Gu L, Cao X | 2014 | Control of agricultural traits by hc-siRNA associated MITEs in rice | https://www.ncbi.nlm.nih.gov/geo/query/acc.cgi?acc=GSE50778 | Publicly available at the NCBI Gene Expression Omnibus (accession no: GSE50778) |
| Zhang QJ, Zhu T, Xia EH, Shi C, Liu YL, Zhang Y, Liu Y, Jiang WK, Zhao YJ, Mao SY, Zhang LP, Huang H, Jiao JY, Xu PZ, Yao QY, Zeng FC, Yang LL, Gao J, Tao DY, Wang YJ, Bennetzen JL, Gao LZ | 2014 | Rapid diversification of five Oryza AA genomes associated with rice adaptation | https://www.ncbi.nlm.nih.gov/bioproject/PRJNA264484 | Publicly available at the NCBI Gene Expression Omnibus (accession no: PRJNA264484) |

| Zhang QJ, Zhu T, Xia EH, Shi C, Liu YL, Zhang Y, Liu Y, Jiang WK, Zhao YJ, Mao SY, Zhang LP, Huang H, Jiao JY, Xu PZ, Yao QY, Zeng FC, Yang LL, Gao J, Tao DY, Wang YJ, Bennetzen JL, Gao LZ | 2014 | Rapid diversification of five Oryza AA genomes associated with rice adaptation | https://www.ncbi.nlm.nih.gov/bioproject/?term=PRJNA264480 | Publicly available at the NCBI Gene Expression Omnibus (accession no: PRJNA264480) |
|---|---|---|---|---|
| Zhang QJ, Zhu T, Xia EH, Shi C, Liu YL, Zhang Y, Liu Y, Jiang WK, Zhao YJ, Mao SY, Zhang LP, Huang H, Jiao JY, Xu PZ, Yao QY, Zeng FC, Yang LL, Gao J, Tao DY, Wang YJ, Bennetzen JL, Gao LZ | 2014 | Rapid diversification of five Oryza AA genomes associated with rice adaptation | https://www.ncbi.nlm.nih.gov/bioproject/?term=PRJNA264485 | Publicly available at the NCBI Gene Expression Omnibus (accession no: PRJNA264485) |
| Zhang F, Xu T, Mao L, Yan S, Chen X, Wu Z, Chen R, Luo X, Xie J, Gao S | 2016 | Genome-wide analysis of Dongxiang wild rice (Oryza rufipogon Griff.) to investigate lost/acquired genes during rice domestication | https://trace.ddbj.nig.ac.jp/DRASearch/study?acc=SRP070627 | Publicly available at the DNA Data Bank of Japan (accession no: SRP070627) |
| Zhai R, Cheng S | 2013 | Transcriptome Analysis of Rice Root Heterosis by RNA-Seq | https://www.ncbi.nlm.nih.gov/geo/query/acc.cgi?acc=GSE41797 | Publicly available at the NCBI Gene Expression Omnibus (accession no: GSE41797) |
| Klepikova AV, Kasianov AS, Gerasimov ES, Logacheva MD, Penin AA | 2016 | A high resolution map of the Arabidopsis thaliana developmental transcriptome based on RNA-seq profiling | https://www.ncbi.nlm.nih.gov/bioproject/?term=PRJNA314076%20 | Publicly available at the NCBI Gene Expression Omnibus (accession no: PRJNA314076) |
| Wu L, Zhang Q, Zhou H, Ni F, Wu X, Qi Y | 2009 | Identification of small RNAs in rice AGO1 complexes and their targets | https://www.ncbi.nlm.nih.gov/geo/query/acc.cgi?acc=GSE18251 | Publicly available at the NCBI Gene Expression Omnibus (accession no: GSE18251) |
| Johnson C, Kasprzewska A, Tennessen K, Fernandes J, Nan G, Walbot V, Sundaresan V, Vance V, Bowman L | 2009 | Endogenous small RNAs of meristematic and a terminally differentiated tissue of rice | https://www.ncbi.nlm.nih.gov/geo/query/acc.cgi?acc=GSE16350 | Publicly available at the NCBI Gene Expression Omnibus (accession no: GSE16350) |
| Wang H, Zhang X, Liu J, Kiba T, Woo J, Ojo T, Hafner M, Tuschl T, Chua N, Wang X | 2011 | Characterization of AGO1-/AGO4-associated smRNAs | https://www.ncbi.nlm.nih.gov/geo/query/acc.cgi?acc=GSE28591 | Publicly available at the NCBI Gene Expression Omnibus (accession no: GSE28591) |

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
