## [Decision Letter]

Thank you for submitting your article "Regulation of rice root development by a retrotransposon acting as a microRNA sponge" for consideration by *eLife*. Your article has been reviewed by two peer reviewers, one of whom, Christian Hardtke, acted as the Reviewing and Senior Editor. The reviewers have opted to remain anonymous.

The reviewers have discussed the reviews with one another and the Reviewing Editor has drafted this decision to help you prepare a revised submission.

As you can see from the detailed reviews below, the reviewers were quite positive about your paper, but would like you to address a number of issues regarding data presentation and documentation before publication. Please see the comments below for details.

Reviewer #1:

Cho and Paszkowski present evidence for microRNA sponge action of a particular transposon in rice. This is a timely subject and reveals another layer of uncommon gene regulation, and is thus interesting for a wide audience. Apparently the manuscript has gone through in depth peer review previously and been transferred to this journal, so the present version is already revised and a response to previous reviews was provided.

Overall, I found the manuscript to be of high quality, and the interpretation of the data credible. Rice is a more difficult system to work with than, say, Arabidopsis, and although I would have liked to see some additional experiments (for example, higher resolution expression analyses), I acknowledge that in the rice system, it is not always easily feasible. For instance, the genetics is in part limit because no complementation lines are provided, but this is offset by the analysis of two independent alleles in each case. A few suggestions for further improvement of the manuscript:

– Statistical tests are not always clear. Which has been used, and why? Please amend figure legends and Materials and methods. In some cases, no test is provided. For example, Figure 1—figure supplement 1: is the difference between expressed and silenced TEs in homology patches really significant? A simple F-test should give the answer. Also, Figure 2, etc. In some figures, error bars are missing? Or not visible because too small (e.g. Figure 4)?

– How was the Arabidopsis expression correlation matrix produced? From published data, following the same procedure as for rice? Please clarify.

– Figure 3: miR171 was not at all detectable in MIKKI O.E. lines? Or value too small? Maybe show with log scale in cases of strong expression differences.

– Regarding miRNA expression: maybe it would be generally more correct to talk about miRNA abundance in the different backgrounds, rather than expression, which most would associate with gene expression in this case (which, as authors show, is not affected in the different backgrounds).

– The whole aspect of differential miR171 regulation in rice versus Arabidopsis or Brachypodium sort of distracts without further investigation. SCL control of miR171 in Arabidopsis has been shown and can remain, but the Brachypodium data are preliminary and not really helpful, could be removed.

– The most important part of the paper is the demonstration that miR171 abundance is increased in the mikki mutant alleles. Could the authors provide small RNA blots to demonstrate that in mikki roots miR171 is specifically up-regulated as compared to a control miRNA? qRT-PCR speaks for it, but given this is a key result, a technically independent verification would be desirable.

Reviewer #2:

This manuscript describes the role of a domesticated rice transposable element in regulating rice development through modulation of miRNA171. The findings are interesting and the data are convincing. As the manuscript already underwent rigorous peer review, I have relatively little to add:

1) A more detailed explanation of the plots in Figure 1—figure supplement 1 would be very helpful. It's not clear what is being correlated, and thus the distinction between TE expression in rice and Arabidopsis does not appear well-supported.

2) Please explain how TE age was estimated in the Materials and methods section (in addition to citing the reference).

3) I believe the authors are not using the term "positive selection" as intended. Positive selection means increased frequency of a new, advantageous mutation in a population. The authors appear to argue that mutations that alter MIKKI structure are disfavored – this is negative selection.

---

## [Author Response]

*Reviewer #1:*

*Cho and Paszkowski present evidence for microRNA sponge action of a particular transposon in rice. This is a timely subject and reveals another layer of uncommon gene regulation, and is thus interesting for a wide audience. Apparently the manuscript has gone through in depth peer review previously and been transferred to this journal, so the present version is already revised and a response to previous reviews was provided.*

*Overall, I found the manuscript to be of high quality, and the interpretation of the data credible. Rice is a more difficult system to work with than, say, Arabidopsis, and although I would have liked to see some additional experiments (for example, higher resolution expression analyses), I acknowledge that in the rice system, it is not always easily feasible. For instance, the genetics is in part limit because no complementation lines are provided, but this is offset by the analysis of two independent alleles in each case. A few suggestions for further improvement of the manuscript:*

*– Statistical tests are not always clear. Which has been used, and why? Please amend figure legends and Materials and methods. In some cases, no test is provided. For example, Figure 1—figure supplement 1: is the difference between expressed and silenced TEs in homology patches really significant? A simple F-test should give the answer. Also, Figure 2, etc. In some figures, error bars are missing? Or not visible because too small (e.g. Figure 4)?*

We performed statistical tests for all our data and now we described them the Materials and methods and mentioned in the corresponding figure legends. Error bars in Figure 4 are indeed very small and are now thickened for better visibility.

*– How was the Arabidopsis expression correlation matrix produced? From published data, following the same procedure as for rice? Please clarify.*

In the figure legend for Figure 1—figure supplement 1, we added more details about how the expression correlation matrix was produced. We also added Supplementary file 3 listing Arabidopsis samples and their dataset sources.

*– Figure 3: miR171 was not at all detectable in MIKKI O.E. lines? Or value too small? Maybe show with log scale in cases of strong expression differences.*

We would like to avoid log scale because the observed differences can be easier related to MIKKI activity and the stoichiometry of its regulatory activity. However, for better display in Figure 3, we broke the y axes and plotted the data in two different scales.

*– Regarding miRNA expression: maybe it would be generally more correct to talk about miRNA abundance in the different backgrounds, rather than expression, which most would associate with gene expression in this case (which, as authors show, is not affected in the different backgrounds).*

Thank you for noticing this, we changed “expression” to “transcript” or “RNA levels” throughout the manuscript.

*– The whole aspect of differential miR171 regulation in rice versus Arabidopsis or Brachypodium sort of distracts without further investigation. SCL control of miR171 in Arabidopsis has been shown and can remain, but the Brachypodium data are preliminary and not really helpful, could be removed.*

We still think that Brachypodium data is of interest because it demonstrates that post-transcriptional regulation of miR171 is not restricted to *Oryza* species. Thus, we would like to leave this part of the manuscript unchanged.

*– The most important part of the paper is the demonstration that miR171 abundance is increased in the mikki mutant alleles. Could the authors provide small RNA blots to demonstrate that in mikki roots miR171 is specifically up-regulated as compared to a control miRNA? qRT-PCR speaks for it, but given this is a key result, a technically independent verification would be desirable.*

We agree with the reviewer’s suggestion and we performed small RNA blot of RNAs extracted from roots of the mutants and wild type, which are consistent with RT-qPCR results. These new data are displayed in Figure 3—figure supplement 2.

*Reviewer #2:*

*This manuscript describes the role of a domesticated rice transposable element in regulating rice development through modulation of miRNA171. The findings are interesting and the data are convincing. As the manuscript already underwent rigorous peer review, I have relatively little to add:*

*1) A more detailed explanation of the plots in Figure 1—figure supplement 1 would be very helpful. It's not clear what is being correlated, and thus the distinction between TE expression in rice and Arabidopsis does not appear well-supported.*

Correlation matrix in Figure 1—figure supplement 1 and B is all-against-all comparison of expression patterns of each TE. Each cluster represents a group of TEs with similar expression profile and more clusters mean higher diversity of TE expression patterns. As answered to the first reviewer, in the revised manuscript we provided additional explanation how the correlation matrix was obtained (legend of Figure 1—figure supplement 1).

*2) Please explain how TE age was estimated in the Materials and methods section (in addition to citing the reference).*

In the revised manuscript we provided detailed explanation in the Materials and methods section how LTR insertion time was estimated (subsection “Age estimation of LTRs”).

*3) I believe the authors are not using the term "positive selection" as intended. Positive selection means increased frequency of a new, advantageous mutation in a population. The authors appear to argue that mutations that alter MIKKI structure are disfavored – this is negative selection.*

The sentence meant to point out that the rearrangement of TE sequences constituting *MIKKI* had selective advantage as demonstrated by conservation of the final *MIKKI* sequence in various rice species. We indeed think that it is example of positive selection but to avoid any misunderstandings we edited the corresponding sentence as follows: “suggesting strong selective advantage of this particular transposon[…]”.